# Effect of Part Size, Displacement Rate, and Aging on Compressive Properties of Elastomeric Parts of Different Unit Cell Topologies Formed by Vat Photopolymerization Additive Manufacturing

**DOI:** 10.3390/polym16223166

**Published:** 2024-11-13

**Authors:** Lindsey B. Bezek, Sushan Nakarmi, Alexander C. Pantea, Jeffery A. Leiding, Nitin P. Daphalapurkar, Kwan-Soo Lee

**Affiliations:** 1Los Alamos National Laboratory, Chemistry Division, P.O. Box 1663, Los Alamos, NM 87545, USA; lbezek@lanl.gov (L.B.B.); apantea@lanl.gov (A.C.P.); 2Los Alamos National Laboratory, Theoretical Division, P.O. Box 1663, Los Alamos, NM 87545, USA; snakarmi@lanl.gov (S.N.);; 3Los Alamos National Laboratory, Materials Physics and Applications Division, P.O. Box 1663, Los Alamos, NM 87545, USA

**Keywords:** additive manufacturing, 3D printing, cellular materials, elastomer, finite element modeling, compression

## Abstract

Due to its ability to achieve geometric complexity at high resolution and low length scales, additive manufacturing (AM) has increasingly been used for fabricating cellular structures (e.g., foams and lattices) for a variety of applications. Specifically, elastomeric cellular structures offer tunability of compliance as well as energy absorption and dissipation characteristics. However, there are limited data available on compression properties for printed elastomeric cellular structures of different designs and testing parameters. In this work, the authors evaluate how unit cell topology, part size, the rate of compression, and aging affect the compressive response of polyurethane-based simple cubic, body-centered, and gyroid structures formed by vat photopolymerization AM. Finite element simulations incorporating hyperelastic and viscoelastic models were used to describe the data, and the simulated results compared well with the experimental data. Of the designs tested, only the parts with the body-centered unit cell exhibited differences in stress–strain responses at different part sizes. Of the compression rates tested, the highest displacement rate (1000 mm/min) often caused stiffer compressive behavior, indicating deviation from the quasi-static assumption and approaching the intermediate rate response. The cellular structures did not change in compression properties across five weeks of aging time, which is desirable for cushioning applications. This work advances knowledge on the structure–property relationships of printed elastomeric cellular materials, which will enable more predictable compressive properties that can be traced to specific unit cell designs.

## 1. Introduction

There has been a lot of recent attention on additive manufacturing (AM) being utilized to fabricate elastomers [1], including for medical applications [2], radio frequency antennas and strain sensors [3], soft robotics [4], and flexible electronics [5]. The tunability of compliance and energy absorption and dissipation in these elastomers for specific applications can be achieved by modifying the elastomeric formation as well as by modifying the part structure through material reduction. AM has excelled at providing ample design flexibility and customizability for fabricating a variety of cellular materials, including stochastic foams, strut-based lattices, and structures derived from skeletal- and shell-based triply periodic minimal surfaces [6,7,8,9]. Furthermore, elastic cellular structures have been demonstrated using a variety of AM techniques, including fused filament fabrication [10,11], direct ink write [12,13,14], material jetting [15,16], powder bed fusion [17], and vat photopolymerization.

Vat photopolymerization, also commonly called stereolithography, is a promising avenue for fabricating advanced elastomers with complex geometries, as demonstrated with printed silicone [18] and latex [19]. Formlabs’ Elastic 50A, a polyurethane-based resin, is one of the more common commercially available elastomeric materials for this process. Elastic 50A has previously been used for biomedical applications [20,21], for spring-like electronics [22], as the matrix material for filled systems [23,24], and for validating frameworks for creating auxetic structures [25,26,27]. Elastic 50A has been utilized to study the compression response for a variety of unit cell topologies [28,29].

In previous work, Nakarmi and co-authors identified buckling and bending responses in Elastic 50A parts consisting of nine different topologies. The authors highlighted the distinct stages of deformation within the compression cycle, which were validated by mesoscale finite element (FE) simulations [29]. Despite this promising work for a better understanding of stress–strain responses and energy-absorbing efficiency in a variety of structures, there were limitations in the explored test parameters that are necessary to study in thorough experimental characterization. These limitations restrict the comprehensive understanding of the structure–property relationships for additively manufactured elastomeric cellular materials. For example, the previous work considered compression experiments and simulations to a maximum of 0.1 MPa; thus, there is an opportunity to explore the effects of larger compressive strains on the deformation response since certain damping and cushioning applications may be subjected to higher loads. Additionally, viscoelastic materials are inherently rate-dependent, motivating the need to investigate this material’s response to different displacement rates in order to broaden the understanding of its performance across a wider spectrum of load-bearing cases. Finally, the effects of part scalability and aging time on the compression response have not been previously tested, which are important factors when considering the durability and reliability of the elastomeric cellular structures in functional applications.

In this work, the authors explored the effects of part size, the displacement rate, and time on the compressive properties of additively manufactured elastomeric cellular structures of varied unit cell topologies. This study utilized FE simulations incorporating the first-order polynomial Mooney–Rivlin hyperelastic model combined with the two-term Prony series for viscoelastic material response to describe the experimental data, which provides insights into the non-linear deformation for each of the cellular structures. The novelty of this work is the exploration of three key elements for foam applications, size scalability, loading rate, and aging, which, to the authors’ knowledge, have not been rigorously studied in additively manufactured elastomeric cellular materials. A better understanding of the compressive response for different part designs and testing parameters will enable predictable, and consequently tailorable, compressive properties for extreme loading conditions and stringent design constraints.

## 2. Materials and Methods

### 2.1. Part Design

Three unit cell topologies were selected for investigation: two strut-based geometries, simple cubic (SC) and body-centered (BC), and one triply periodic minimal surface, gyroid (G), as illustrated in Figure 1a. Solid (S) structures were also generated as a bulk control. For all cellular structures, the unit cell size was 5 mm, and parts were designed to have a 20% volume fraction of the material. To test different part sizes, the unit cells were structurally patterned as either 3 × 3 × 3, 6 × 6 × 6, or 12 × 12 × 12, as shown in Figure 1b.

Computer-aided design (CAD) models of these structures were generated using the parametric design platform in Grasshopper (Rhinoceros 7 plugin) [30]. The methodology for digitally creating these structures has been communicated in a previous study [29]. The generated models were then converted to stereolithography (STL) formats for printing. The STLs were additionally used to create the FE models for compression simulation.

### 2.2. Materials and Manufacturing

Parts were fabricated using Formlabs’ Elastic 50A on a Formlabs Form 3+ (Somerville, MA, USA) with 100 µm layers and no supports. For all structures except the 12 × 12 × 12 BC and G, the early layer exposure was reduced from the default early layer exposure (671 mJ/cm^2^) to 350 mJ/cm^2^ to mitigate overprinting of the initial layers. The largest BC and G structures required the default early layer exposure to ensure they would remain adhered to the build platform over the duration of the print.

Once finished printing, parts were removed from the build plate within 1 h. Parts were submerged in isopropyl alcohol (IPA) for 10 min, briefly rinsed in deionized water, submerged again in IPA for 10 min, and again briefly rinsed in deionized water. During these cleaning steps, parts were gently agitated to ensure all residual liquid was removed. Parts were then submerged in deionized water and cured under ultraviolet (UV) light in the Formlabs Form Cure for 5 min, which removed remaining tackiness from the parts. Parts were dried with compressed air followed by being heated for 20 min at 60 °C. Parts were tested at least 24 h later. Compared to the post-processing protocol for the same material in a previous study [29], which included three days of submersion in IPA and no UV curing, the protocol for the current study was modified to more closely reflect the suggested post-processing by the supplier [31,32]. A brief study highlighting the differences in mechanical performance between these procedures is provided in Appendix B.

### 2.3. Mechanical Testing

Compression tests of the 3 × 3 × 3 and 6 × 6 × 6 structures were performed on an Instron 3343 with a 1 kN load cell. This mechanical testing system was too small to accommodate the 12 × 12 × 12 structures, which were tested on an Instron 5969 with a 50 kN load cell. All parts were oriented such that the print direction (*Z*-axis) corresponded to the compression axis (i.e., the part’s first layer was on the bottom compression platen). The top and bottom surfaces of all parts were directly in contact with the steel platen, and the tests did not use any lubrication or other solvents, thereby leaving the interface elastomer–steel frictional condition unaltered. For the investigation of different part sizes, each part was compressed at a rate of 80%/min to a maximum strain of 80% over three cycles, and data were collected during the final cycle. For the investigation of different compression rates, the 6 × 6 × 6 parts were displaced either at 12, 120, or 1000 mm/min in three cycles to 0.5 MPa, and data were taken on the last cycle. For the investigation of aging effects, the 6 × 6 × 6 parts were compressed at 12 mm/min in three cycles to 0.5 MPa weekly for five additional weeks. These parts were stored in a dark environment (~20–25 °C, ~15–25% RH) over the duration of the test. A summary of the variables explored in this study can be found in Table 1. For consistency in data analysis, the zero reference (or origin) point for recording the nominal stress–strain data was chosen at a pre-stress point of 0.001.

### 2.4. Simulations

Advanced non-linear finite element methods (FEMs) from SIMULIA’s Abaqus [33] were used to model and understand the mechanical behavior of the three cellular structures. First, the models were discretized with all hexahedral elements to create high-quality meshed structures. Mesh discretization was accomplished using the overlay grid method called ‘*Sculpt*’ in Cubit version 16.06.0. [34]. A grid size of 0.15 mm was selected based on a reasonable demonstration of the convergence of the overall stress–strain response from previous work [29]. The scaled Jacobian of the generated mesh was also evaluated and found to be above 0.2. This is a satisfactory threshold for a high-quality mesh in order to accomplish the FE simulations precluding excessive mesh distortion and degeneration errors commonly encountered in large deformation analysis.

For the material property definition, the simulations used the same hyperelastic and viscoelastic models that previously generated simulations that aligned well with experimental results for the same material and unit cell topologies [29]. The models were calibrated from tensile, compression, and relaxation tests using methods similar to those in previous work [29]. The first-order polynomial Mooney–Rivlin hyperelastic model was chosen based on the accuracy and inherent stability at all strains. The strain energy potential (U) for the Mooney–Rivlin model is given by
(1)$$U=C_{10}\left({\bar{I}}_{1}-3\right)+C_{01}\left({\bar{I}}_{2}-3\right)+\frac{1}{D_{1}}{\left(J_{el}-1\right)}^{2}$$

Here, $C_{10}$, $C_{01}$, and $D_{1}$ are the material constants that require calibration. ${\bar{I}}_{1}$ and ${\bar{I}}_{2}$ are the first and the second invariants of the deviatoric component of the left Cauchy–Green deformation tensor, respectively, and $J_{el}$ is the Jacobian of the deformation gradient tensor. The values of the material constants obtained from the calibration process are $C_{10}=550829.02\;\mathrm{Pa}$, $C_{01}=84650.30\;\mathrm{Pa}$, and $D_{1}=6.38\times 10^{-8}\;{\mathrm{Pa}}^{-1}$.

The viscoelastic material characterization utilized the two-term Prony series that was calibrated from a relaxation test where the specimen was strained to a nominal strain of 21% and held constant for 3.5 h. The time-dependent normalized shear modulus ($g_{R}$) in the form of the Prony series is
(2)$$g_{R}\left(t\right)=1-\overset{2}{\underset{i=1}{\sum }}\bar{g_{i}}\left(1-e^{-\frac{t}{\tau_{i}}}\right).$$

Here, $g_{i}$ and $\tau_{i}$ are the normalized shear modulus and relaxation time, respectively, for the $i^{th}$ term in the series. The calibrated model parameters are $g_{1}=5.59\times 10^{-2}$, $\tau_{1}=35.77\;s$, $g_{2}=1.56\times 10^{-2}$, and $\tau_{2}=2930.7\;s$.

Simulations under unconfined compression were performed using Abaqus/Explicit, which essentially solves the elastodynamic governing equations. The discretized FE models were positioned in between two rigid platens with platen motion along the *Z*-axis. The lower platen was fixed by imposing zero displacements and rotations on all six degrees of freedom. Then, a constant velocity was assigned to the top platen along the negative *Z*-axis (simulating downward platen motion). The ‘*general contact*’ algorithm [33] was used, which handles the contact between all the components of the cellular structure and the platen. The normal and tangential behavior of the contact property was defined with the default ‘*hard*’ contact pressure–overclosure relationship and ‘*penalty*’ friction formulation with a friction coefficient of 0.75. 

## 3. Results

The SC, BC, and G structures experience three distinct deformation responses, consistent with what was observed in previous work [29]. As shown in Figure 2a, the SC structure undergoes global structural buckling, for which the entire part buckles as a unit [16]. Alternatively, the BC structure experiences bending (Figure 2b), where the struts accommodate part deformation through strut rotations instead of buckling [29]. Finally, the G structure (Figure 2c) experiences buckling in individual unit cells. This local buckling appears to occur across planes of unit cells normal to the compression axis, while other planes of unit cells remain undeformed until a later stage. This tiered collapsing was also observed in a previous study when the volume fraction was at or below 20% for thermoplastic gyroids [11].

The subsequent sections show the compressive responses for the parts with different sizes (Section 3.1), testing rates (Section 3.2), and aging conditions (Section 3.3). All part dimensions are provided in Table A1 in Appendix C. Videos of the compression of the 3 × 3 × 3 structures are provided as Appendix A.

### 3.1. Effect of Part Size

The experimental compressive responses of the SC structures of different part sizes are shown in Figure 3a along with the model outputs. For better visibility of the comparisons between the simulations and the experiments for each part size, the data are replotted in Figure 3d,g, and j for the 3 × 3 × 3, 6 × 6 × 6, and 12 × 12 × 12 cases, respectively. For this unit cell topology, there are minimal differences in the stress–strain profiles between the different part sizes. The close alignment between corresponding simulations (indicated in the figure legends with *Model*) and experiments demonstrates that the FE model is appropriate for describing the SC compressive response.

For the BC structures, which are shown in Figure 3b,e,h,k, the compressive stresses remain lower than those of the SC structures during the initial bending mechanism. At ~50–60% strain, the deformation transitions to a densification regime, for which there is a distinguished shift in the stress–strain behavior, which becomes more pronounced with increased part scaling. While the 3 × 3 × 3 model (Figure 3e) aligns well with the experiments, the 6 × 6 × 6 model (Figure 3h) and the 12 × 12 × 12 model (Figure 3k) both predict that the shift occurs at a higher strain and stress. Of the three unit cell topologies evaluated, this shift in the stress–strain plot is unique to the BC structure. Interestingly, this shift is not present in compression tests of rigid BC structures [35,36,37]; consequently, it could be a result of stretching-dominated characteristics during densification. Stretching-dominated behavior generally consists of an initial peak in the stress profile [36,37]. It is possible that at the higher strains, the cellular structure’s response is no longer bending-dominated, and the elastic struts are able to stretch to accommodate the increased strain. This would also explain how the 3 × 3 × 3 structure has a less pronounced shift since the cellular structure consists of less material. The location of the shift may be indicative of the moment when multiple layers collapse [38]; however, further studies are needed to investigate how to better predict the precise location and magnitude of this shift.

The results for the G structure are provided in Figure 3c,f,i,l. Although the G structure experiences local unit cell buckling compared to the SC structure’s global buckling, the stress–strain profile is relatively similar in shape, with an initial linear elastic region followed by a stress plateau until the densification stage. The FE models accurately reflect the experimental compressive response.

### 3.2. Effect of Displacement Rate

Figure 4 displays the compressive responses of the 6 × 6 × 6 cellular structures when subjected to displacement rates of different orders of magnitude, and Table 2 reports the corresponding average values. For elastomers, the rate of deformation affects the balance between elastic and viscous behavior and consequently the mechanical response [39]. At higher displacement rates, it is expected that structures will have higher stiffnesses because there is insufficient time for viscous energy dissipation. The SC stress–strain response has overlapping trends for the initial linear elastic and plateau regions. Increasing strain rate very slightly decreases the strain required to reach 0.5 MPa, but parts only vary between 75.7% and 78.5% strain, while rate dependency is more pronounced for the other topologies. The 12 and 120 mm/min trends generally overlap for the BC, G, and S structures in the regimes other than densification. In the densification regime, the stress–strain curves generally demonstrate an insignificant stiffer response with the increasing rate, with the most pronounced and significant effect for the BC structure. For the G and S structures at 120 mm/min, the compressive strains at 0.5 MPa are slightly less than those of the 12 mm/min case, but when considering the sample standard deviations (Table 2), these values are comparable. The 1000 mm/min test cases have noticeably stiffer responses for the BC, G, and S structures. With this being the maximum testing rate for the given frame, it is possible that the resolution of the signal was limited by the 0.02 s sampling frequency. The S structure is the only case showing a rate dependency below ~4% strain, where the 1000 mm/min test case experiences lower compressive stress compared to those at the lower displacement rates. This viscoelastic effect is likely only present in the bulk material since the SC, BC, and G structures have designed free volume that can accommodate the initial response to the higher displacement rate.

### 3.3. Effect of Aging

The 6 × 6 × 6 cellular structures were retested weekly at 12 mm/min for five subsequent weeks. The parts did not show any macroscopic indication of damaged struts from the repeated testing over the duration of the study. As shown in Figure 5, the SC, BC, and G structures maintained ~80% strain at 0.5 MPa, and the S structure maintained ~10% strain. While some variation exists across weeks, such as a slight drop between BC weeks 3 and 4, these shifts are considered negligible when considering the variation from the sample standard deviation, and they could be attributed to human error from taking measurements on an elastomer.

The lack of major changes in compressive behavior across the five additional weeks shows promise for this material system serving in cushioning applications; however, a more extensive aging study would be necessary to evaluate this material’s longer-term durability. Hong and co-authors used the Arrhenius equation to predict the lifespan of a polyurethane elastomer when considering the measured tensile strength and elongation at break values from a heat-aging study, and the estimated lifetime based on tensile strength was 12.2 years at room temperature [40]. Although accelerated thermal aging combined with the Arrhenius method is recognized as one of the more reliable approaches for predicting the lifetime of elastomers, there are several mechanisms of degradation for elastomers (e.g., chemical, mechanical, and environmental factors) that should also be considered in future studies to effectively predict long-term performance capability [41].

This study has evaluated mechanical properties at room temperature and standard humidity conditions. However, testing at other temperatures and humidities would be useful when considering practical applications that may subject these materials to other environmental conditions. Kanyanta and co-authors showed that increasing temperature and humidity levels softens an ether-based polyurethane elastomer, evaluated through tensile studies [42]. Opreni and co-authors also saw effects of temperature and relative humidity on the compression response of elastomeric polyurethane foams, finding that time–temperature–superposition principles could be applied to predict the viscoelastic responses across broad temperature and humidity ranges [43]. Extending the study to other temperatures and humidities would be of interest in future work.

As with many photopolymer systems, it is assumed that the material is sensitive to additional UV exposure [44]. If there is residual monomer, additional UV exposure could lead to higher crosslink density, resulting in a stiffer response, as seen in Appendix B (Figure A1). At some point, prolonged UV exposure would lead to degradation. Polyurethane acrylates could be prone to discoloration and chemical modification due to UV aging [45]. Although investigating different UV exposures is outside of the scope of this study, it would be valuable to conduct this experiment in the future to identify the connection between the degree of conversion and mechanical properties.

## 4. Conclusions

AM, which enables the creation of complex elastomeric parts, shows potential for fabricating parts requiring tunable compliance and damping across a broad spectrum of applications. This study advances knowledge on the compressive performance of cellular structures formed by vat photopolymerization by examining the effects of part size, the displacement rate, and aging over five weeks. Uniaxial compression tests were conducted on parts consisting of three different unit cell topologies, as well as solid cubes, and simulations integrating hyperelastic and viscoelastic models were developed to compare with the experimental findings. The key findings from this study include the following:In general, the results from the FE simulations exhibited excellent comparison to experimental data when qualitatively and quantitatively describing the compressive responses of the SC, BC, and G structures at different part sizes.Across the part sizes tested, the experiments and simulations consistently showed there were no appreciable differences in stress–strain responses except for the BC structures, which experienced shifts in the densification region.The BC, G, and S structures experienced stiffer stress–strain responses at the highest displacement rate tested.The tested structures did not show substantial changes in the compressive response over the duration of five weeks of aging time.

Improving the understanding of structure–property relationships for additively manufactured elastomers guides material design for achieving desired mechanical responses. Specifically, the experimental data and simulations provide a foundation to predict and control compressive properties, where future work can utilize methods such as topology optimization to inform unit cell designs based on emerging part requirements.

Although this study was limited to one feedstock material and volume fraction, the validated simulation could be applied to predict compressive responses in other volume fractions or topologies. Furthermore, the simulation could be applied to other materials when calibrated with updated material properties. Although the materials were tested across five weeks, it would be beneficial to conduct further evaluation across a longer duration in the event of polymer degradation. Fatigue testing could also supplement this analysis. Another limitation of this study was that the displacement rates tested, though varied across two orders of magnitude, were all within the quasi-static testing range. There would be interest in future work extending to higher strain rate testing to further increase knowledge about elastomeric cellular material performance for a more extensive application space.

## Figures and Tables

**Figure 1 polymers-16-03166-f001:**
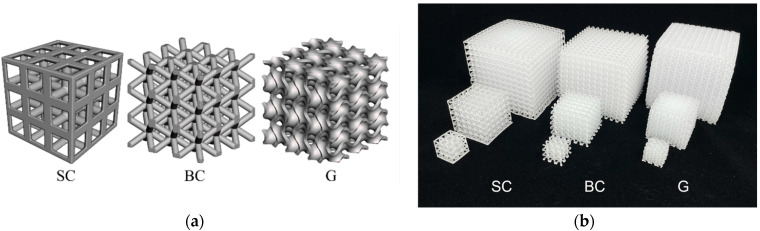
(**a**) Digital renderings of the simple cubic (SC), body-centered (BC), and gyroid (G) cellular structures with 5 mm unit cells patterned into cubes with side lengths made up of 3 unit cells; (**b**) samples of printed SC, BC, and G cellular structures with 5 mm sized unit cells and side lengths made up of 3, 6, and 12 unit cells.

**Figure 2 polymers-16-03166-f002:**
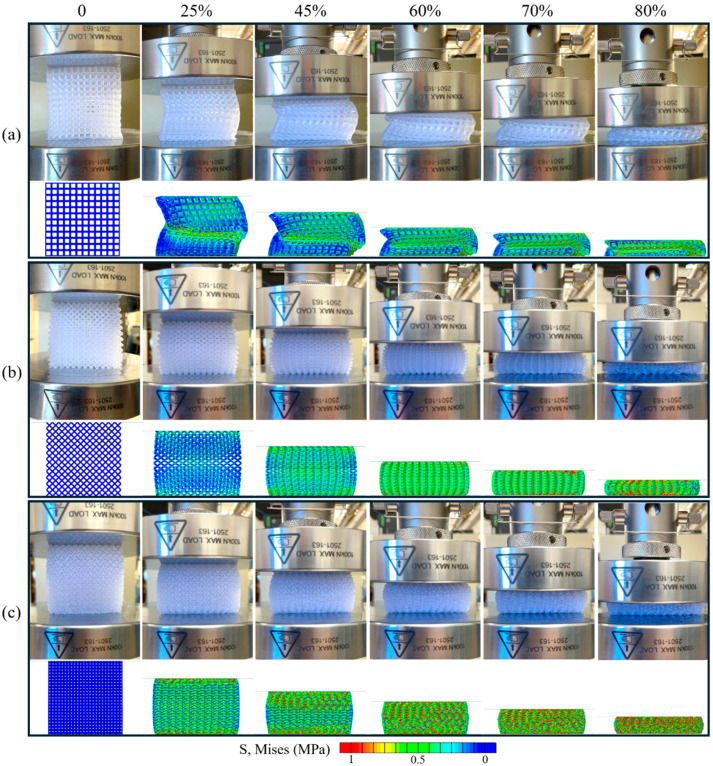
Evolution of compression response for the (**a**) SC, (**b**) BC, and (**c**) G structures with 12 × 12 × 12 unit cells strained to 80%. For individual structures, top rows are snapshots from experiments and the bottom rows are corresponding snapshots from simulations. The SC structure experiences global structural buckling, while the BC and G parts deform more uniformly according to the unit cell design.

**Figure 3 polymers-16-03166-f003:**
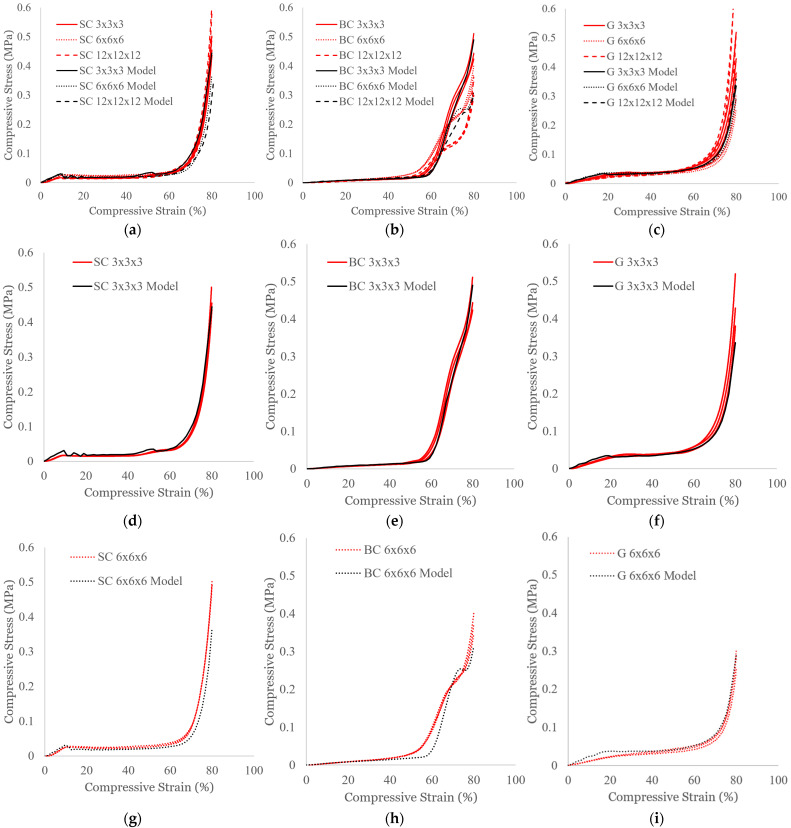
Compressive stress–strain profiles for (**a**) SC, (**b**) BC, and (**c**) G structures of different part sizes. The data are replotted separately for (**d**–**f**) 3 × 3 × 3, (**g**–**i**) 6 × 6 × 6, and (**j**–**l**) 12 × 12 × 12 part sizes for SC, BC, and G structures, respectively. Simulated results are indicated using ‘Model’ and plotted using black lines. Experimental results (n = 3) are plotted with red lines.

**Figure 4 polymers-16-03166-f004:**
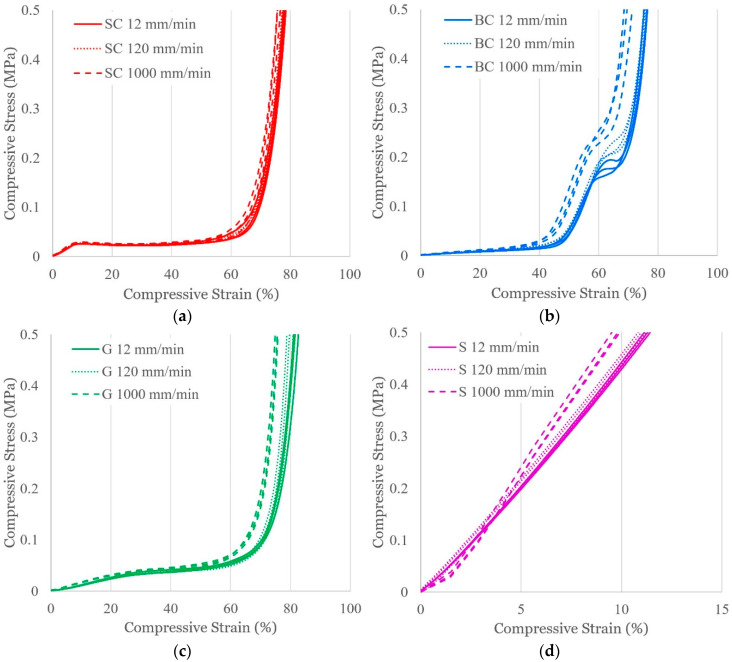
Compressive stress–strain profiles for (**a**) SC, (**b**) BC, (**c**) G, and (**d**) S structures when subjected to varying displacement rates. n = 3.

**Figure 5 polymers-16-03166-f005:**
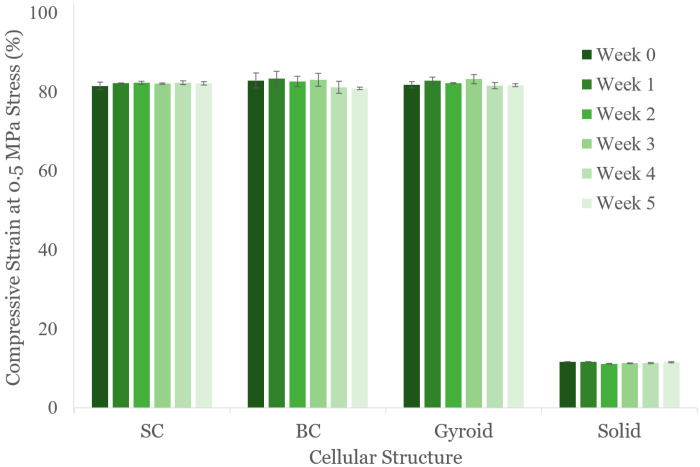
Compressive strains of the cellular structures tested up to five weeks after printing. Error bars represent one sample standard deviation across three samples.

**Table 1 polymers-16-03166-t001:** Summary of testing parameters for each of the experiments in this study.

Experiment	Unit Cell Topology	Part Size (Unit Cells per Length)	Testing Rate	Additional Studies
Part size	SC, BC, or G	3 × 3 × 3, 6 × 6 × 6, or 12 × 12 × 12	80%/min to 80%	Compared to simulations
Displacement rate	SC, BC, G, or S	6 × 6 × 6	12, 120, or 1000 mm/min to 0.5 MPa	N/A
Aging	SC, BC, G, or S	6 × 6 × 6	12 mm/min to 0.5 MPa	Tested weekly for five weeks

**Table 2 polymers-16-03166-t002:** Averages (and one sample standard deviation) of compressive strains required to reach 0.5 MPa compressive stress for each of the cellular structures and displacement rates.

	SC	BC	G	S
12 mm/min	78.04 ± 0.41	75.98 ± 0.63	81.72 ± 0.80	11.29 ± 0.15
120 mm/min	77.35 ± 0.18	75.65 ± 0.44	79.75 ± 1.04	11.01 ± 0.20
1000 mm/min	75.99 ± 0.16	69.88 ± 1.33	75.48 ± 0.45	9.78 ± 0.20

## Data Availability

Data are available upon request.

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
