# Peer review of "Effect of Part Size, Displacement Rate, and Aging on Compressive Properties of Elastomeric Parts of Different Unit Cell Topologies Formed by Vat Photopolymerization Additive Manufacturing"

_polymers, 2024, doi:10.3390/polym16223166_

Round 1

Reviewer 1 Report

Comments and Suggestions for Authors

The manuscript with the title "Effect of part size, displacement rate, and aging on compressive 2 properties of elastomeric parts of different unit cell topologies 3 formed by vat photopolymerization additive manufacturing"

Manuscript ID: polymers-3290263-peer-review-v1-1

Dear Authors,

In the beginning, I would like to express words of my appreciation for the idea and effort put into conducting research and writing the manuscript recommended to me for review.

General Comments:

The article investigates the compressive properties of elastomeric cellular structures produced using vat photopolymerization (3D printing). It examines how part size, displacement rate, and aging affect the stress-strain response of different unit cell topologies, specifically simple cubic, body-centred, and gyroid structures. Finite element (FE) simulations were employed to model and interpret the mechanical response of these cellular structures under varying conditions. This research is valuable for developing advanced elastomeric components with tuneable mechanical properties for various engineering applications, such as cushioning and energy absorption. The study provides a comprehensive experimental setup with clear comparisons across multiple parameters. The rigorous testing of three unit cell types across varied part sizes, strain rates, and aging intervals offers valuable insights into the structural behaviour of these elastomeric materials.

I believed that the use of FE simulations and their comparison with experimental data strengthen the study’s conclusions by providing a detailed view of the material properties and allowing for reliable predictions in other configurations.

Specific Comments:

The work is appropriate for the journal, but there are some observations that must be addressed before the manuscript can be accepted.

1) Although the results for aging effects are promising, the discussion lacks detail regarding long-term implications. Including a more comprehensive analysis or referencing additional research on similar materials' aging behavior would enhance the practical relevance of this study.

2)The article could benefit from a more detailed description of how the FE model parameters were selected for different topologies. Clarifying the reasons behind specific model choices for each topology would reinforce the reliability of the simulations.

3) How do the mechanical properties of elastomers change under variable temperature or humidity conditions? The article does not discuss the impact of environmental factors, such as temperature or humidity, on compression and other mechanical properties of elastomers.

4) What are the effects of prolonged UV exposure on the studied elastomeric structures?

Final State of the review:

The work presented for review shows the exciting work of authors that was put into this manuscript. The article meets the criteria set for authors to be published in POLYMERS.

Therefore, I ask the Editor to recommend this work for publication after MINOR REVISION.

Author Response

The authors thank the reviewer for their time and insightful feedback.  The manuscript has been revised to address the comments.

1) Although the results for aging effects are promising, the discussion lacks detail regarding long-term implications. Including a more comprehensive analysis or referencing additional research on similar materials' aging behavior would enhance the practical relevance of this study.

The following has been added to Section 3.3:

Hong and co-authors use the Arrhenius equation to predict the lifespan of a polyurethane elastomer when considering measured tensile strength and elongation at break values from a heat-aging study, and the estimated lifetime based on tensile strength was 12.2 years at room temperature [40].  Although accelerated thermal aging combined with the Arrhenius method is recognized as one of the more reliable approaches for predicting the lifetime of elastomers, there are several mechanisms of degradation for elastomers (e.g., chemical, mechanical, and environmental factors) that should also be considered in future studies to effectively predict long-term performance capability [41].

[40] S. Hong et al., “Molecular degradation mechanism of segmented polyurethane and life prediction through accelerated aging test,” Polym. Test., vol. 124, p. 108086, Jul. 2023, doi: 10.1016/j.polymertesting.2023.108086.

[41] M. Tayefi, M. Eesaee, M. Hassanipour, S. Elkoun, E. David, and P. Nguyen-Tri, “Recent progress in the accelerated aging and lifetime prediction of elastomers : A review,” Polym. Degrad. Stab., vol. 214, p. 110379, Aug. 2023, doi: 10.1016/j.polymdegradstab.2023.110379.

2) The article could benefit from a more detailed description of how the FE model parameters were selected for different topologies. Clarifying the reasons behind specific model choices for each topology would reinforce the reliability of the simulations.

In Section 2.4, the authors have added clarification that the hyperelastic and viscoelastic models were selected based on previous work (Nakarmi 2024), where simulations based off these models aligned well with experimental results for the same material and unit cell topologies.  In Nakarmi’s study, several hyperelastic models were evaluated, and the strongest fit was selected.

3) How do the mechanical properties of elastomers change under variable temperature or humidity conditions? The article does not discuss the impact of environmental factors, such as temperature or humidity, on compression and other mechanical properties of elastomers.

The following has been added to Section 3.3:

This study has evaluated mechanical properties at room temperature and standard humidity conditions.  However, testing at other temperatures and humidities would be useful when considering practical applications that may subject these materials to other environmental conditions.  Kanyanta and co-authors show that increasing temperature and humidity levels softens an ether-based polyurethane elastomer, evaluated through tensile studies [42].  Opreni and co-authors also saw effects of temperature and relative humidity on the compression response of elastomeric polyurethane foams, finding that time-temperature-superposition principles could be applied to predict the viscoelastic responses across broad temperature and humidity ranges [43].  Extending the study to other temperatures and humidities would be of interest in future work.

[42] V. Kanyanta and A. Ivankovic, “Mechanical characterisation of polyurethane elastomer for biomedical applications,” J. Mech. Behav. Biomed. Mater., vol. 3, no. 1, pp. 51–62, Jan. 2010, doi: 10.1016/j.jmbbm.2009.03.005.

[43] A. Opreni, S. Mariani, M. Dossi, and M. Brennan, “Combined effects of temperature and humidity on the mechanical properties of polyurethane foams,” J. Rheol., vol. 64, no. 1, pp. 161–176, Jan. 2020, doi: 10.1122/1.5094849.

4) What are the effects of prolonged UV exposure on the studied elastomeric structures?

The following has been added to Section 3.3:

As with many photopolymer systems, it is assumed that the material is sensitive to additional UV exposure [44].   If there is residual monomer, additional UV exposure could lead to higher crosslink density, resulting in a stiffer response, as seen in Appendix A.  At some point, prolonged UV exposure would lead to degradation.  Polyurethane acrylates could be prone to discoloration and chemical modification due to UV aging [45].  Although investigating different UV exposures is outside of the scope of this study, it would be valuable to conduct this experiment in the future to identify the connection between degree of conversion and mechanical properties.

[44] L. B. Bezek and C. B. Williams, “Process-structure-property effects of ultraviolet curing in multi-material jetting additive manufacturing,” Addit. Manuf., vol. 73, p. 103640, Jul. 2023, doi: 10.1016/j.addma.2023.103640.

[2] C. Decker, K. Moussa, and T. Bendaikha, “Photodegradation of UV‐cured coatings II. Polyurethane–acrylate networks,” J. Polym. Sci. Part Polym. Chem., vol. 29, no. 5, pp. 739–747, Apr. 1991, doi: 10.1002/pola.1991.080290516

Reviewer 2 Report

Comments and Suggestions for Authors

The article by Lindsey B. Bezek, Sushan Nakarmi, Alexander C. Pantea, Jeffery A. Leiding, Nitin P. Daphalapurkar and Kwan-Soo Lee presents a study the physical and mechanical properties of polymer samples of various topologies with 80% free volume created by vat photopolymerization. The work is performed at a good experimental level and combines experimental data with modeling of the processes occurring during physical and mechanical action on the polymer.

There are two small comments on the work:

1) The authors provide detailed experimental protocols for creating polymer samples, but do not indicate such material characteristics as monomer conversion, which is associated with the physical and mechanical properties of the polymer. UV curing in post processing condition also changes the properties of the polymer due to additional post-polymerization. Therefore, this characteristic is necessary for the correct interpretation of the results obtained.. I recommend adding these experimental data.

2) The authors used very mild conditions (darkness, 25 ). Significant changes in the physical and mechanical properties of polymers over such a short period of time under mild conditions can only occur when using decomposable polymers, which the polymer resin used does not belong to. I recommend paying attention to aging experiments in the future and conducting them not only under mild conditions.

The work can be recommended for publication after these comments are eliminated.

Author Response

The authors thank the reviewer for their time and insightful feedback.  The manuscript has been revised to address the comments.

1) The authors provide detailed experimental protocols for creating polymer samples, but do not indicate such material characteristics as monomer conversion, which is associated with the physical and mechanical properties of the polymer. UV curing in post processing condition also changes the properties of the polymer due to additional post-polymerization. Therefore, this characteristic is necessary for the correct interpretation of the results obtained. I recommend adding these experimental data.

The authors agree that consideration of UV effects as they relate to monomer conversion is important.  Although this was not calculated in this study, the authors feel that the detailed methodology enables replication of the UV conditions used in this study.  In Section 2.2, the authors have added the detail that the post-curing was conducted in a Formlabs Form Cure.  Additionally, the following discussion has been added to Section 3.3:

As with many photopolymer systems, it is assumed that the material is sensitive to additional UV exposure [44].   If there is residual monomer, additional UV exposure could lead to higher crosslink density, resulting in a stiffer response, as seen in Appendix A.  At some point, prolonged UV exposure would lead to degradation.  Polyurethane acrylates could be prone to discoloration and chemical modification due to UV aging [45].  Although investigating different UV exposures is outside of the scope of this study, it would be valuable to conduct this experiment in the future to identify the connection between degree of conversion and mechanical properties.

[44] L. B. Bezek and C. B. Williams, “Process-structure-property effects of ultraviolet curing in multi-material jetting additive manufacturing,” Addit. Manuf., vol. 73, p. 103640, Jul. 2023, doi: 10.1016/j.addma.2023.103640.

[45] C. Decker, K. Moussa, and T. Bendaikha, “Photodegradation of UV‐cured coatings II. Polyurethane–acrylate networks,” J. Polym. Sci. Part Polym. Chem., vol. 29, no. 5, pp. 739–747, Apr. 1991, doi: 10.1002/pola.1991.080290516

2) The authors used very mild conditions (darkness, 25 ℃). Significant changes in the physical and mechanical properties of polymers over such a short period of time under mild conditions can only occur when using decomposable polymers, which the polymer resin used does not belong to. I recommend paying attention to aging experiments in the future and conducting them not only under mild conditions.

The following has been added to Section 3.3:

This study has evaluated mechanical properties at room temperature and standard humidity conditions.  However, testing at other temperatures and humidities would be useful when considering practical applications that may subject these materials to other environmental conditions.  Kanyanta and co-authors show that increasing temperature and humidity levels softens an ether-based polyurethane elastomer, evaluated through tensile studies [42].  Opreni and co-authors also saw effects of temperature and relative humidity on the compression response of elastomeric polyurethane foams, finding that time-temperature-superposition principles could be applied to predict the viscoelastic responses across broad temperature and humidity ranges [43].  Extending the study to other temperatures and humidities would be of interest in future work.

[42] V. Kanyanta and A. Ivankovic, “Mechanical characterisation of polyurethane elastomer for biomedical applications,” J. Mech. Behav. Biomed. Mater., vol. 3, no. 1, pp. 51–62, Jan. 2010, doi: 10.1016/j.jmbbm.2009.03.005.

[43] A. Opreni, S. Mariani, M. Dossi, and M. Brennan, “Combined effects of temperature and humidity on the mechanical properties of polyurethane foams,” J. Rheol., vol. 64, no. 1, pp. 161–176, Jan. 2020, doi: 10.1122/1.5094849.